# Central American and Caribbean Consensus Document for the Optimal Management of Oral Anticoagulation in Patients with Non-Valvular Atrial Fibrillation Endorsed by the Central American and Caribbean Society of Arterial Hypertension and Cardiovascular Prevention

**DOI:** 10.3390/jcm13020314

**Published:** 2024-01-05

**Authors:** Fernando Wyss, Vivencio Barrios, Máxima Méndez, Samuel Ramos, Ángel Gonzalez, Héctor Ortiz, Marco Rodas Díaz, Gabriela Castillo, Daniel Quesada, Carlos Enrique Franco, Jaime Ventura, Emilio Peralta López, Francisco Somoza, Ariel Arguello Montealegre, Daniel Meneses, Daniel Pichel, Osiris Valdez

**Affiliations:** 1Cardiovascular Services and Technology of Guatemala, CARDIOSOLUTIONS, Central American and Caribbean Society of Arterial Hypertension and Cardiovascular Prevention, Guatemala City 01010, Guatemala; 2Cardiology Department, University Hospital Ramon y Cajal, School of Medicine, Alcala University, 28034 Madrid, Spain; 3Cardiologist-Internal Medicine, Medicine Autonomous University, Santo Domingo 10105, Dominican Republic; drammendez18@gmail.com; 4Cardiometabolic Unity, The Hub Innovation and Investigation of the Iberoamerican University, UNIBE, Santo Domingo 10203, Dominican Republic; 5Lipid Master National Lipid Association, Jacksonville, FL 32216, USA; 6Cardiology Department, Presidente Estrella Ureña Hospital, Santiago de los Caballeros, Salvador B. Gautier Hospital, Santo Domingo 10514, Dominican Republic; drsamuelramosu@yahoo.com; 7Cardiology Department, Hospiten Santo Domingo, Autónoma de Santo Domingo University, Santo Domingo 1355, Dominican Republic; angel_cardiologia@hotmail.com; 8American and Caribbean Society of Arterial Hypertension and Cardiovascular Prevention, Guatemala City 01010, Guatemala; osirisvaldez52@gmail.com; 9Cardiology Department, Herrera Llerandi Hospital, Guatemala City 01010, Guatemala; isaac214@gmail.com; 10Guatemalan Association of Cardiology, Cardiovascular Surgery Unit of Guatemala (UNICAR), Guatemala City 01010, Guatemala; cardioguate@gmail.com; 11Cardiology Department, Max Peralta Hospital, Costa Rica University Cartago, Cartago 30101, Costa Rica; gcastillocr@yahoo.com; 12Cardiology Department, Hospital San Vicente de Paul, Universidad de Costa Rica, Heredia 40101, Costa Rica; drdanielquesada@gmail.com; 13Cardiology Department, Surgical and Oncological Medical Hospital, Instituto Salvadoreño del Seguro Social, Escalon Medical Center, San Salvador 1101, El Salvador; carlosenriquefranco@gmail.com; 14Cardiology Department, Instituto Salvadoreño del Seguro Social, San Salvador 1101, El Salvador; jrventura@yahoo.com; 15Cardiology Department, Instituto Nacional Cardiopulmonar, Tegucigalpa 11101, Honduras; samaelperalta@hotmail.com; 16Cardiology Department, CEMESA Hospital, San Pedro Sula 21102, Honduras; frsomozaa05@gmail.com; 17Cardiology Department, Vivían Pellas Hospital, Managua 11044, Nicaragua; dr.arguellom@gmail.com; 18Cardiology Department, Del Valle Cardiologic Clinique, Nacional Autónoma of Nicaragua University, Managua 14145, Nicaragua; jdmeneses@gmail.com; 19Cardiology Department, Paitilla Hospital, University of Panamá, Paitilla 06001, Panama; d.r.pichel@mac.com; 20Cardiology Department, Central Romana Hospital, La Romana 22000, Dominican Republic

**Keywords:** anticoagulation, atrial fibrillation, Vitamin K antagonists, direct oral anticoagulants, Central American and Caribbean, access

## Abstract

Atrial fibrillation (AF) is the most common arrhythmia in adults. Prevention of the ischaemic risk with oral anticoagulants (OACs) is widely recommended, and current clinical guidelines recommend direct oral anticoagulants (DOACs) as preference therapy for stroke prevention. However, there are currently no clinical practice guidelines or recommendation documents on the optimal management of OACs in patients with AF that specifically address and adapt to the Central American and Caribbean context. The aim of this *Delphi*-like study is to respond to doubts that may arise in the management of OACs in patients with non-valvular AF in this geographical area. A consensus project was performed on the basis of a systematic review of the literature, a recommended ADOLOPMENT-like approach, and the application of a two-round *Delphi* survey. In the first round, 31 recommendations were evaluated and 30 reached consensus, of which, 10 unanimously agreed. The study assessed expert opinions in a wide variety of contextualized recommendations for the optimal management of DOACs in patients with non-valvular atrial fibrillation (NVAF). There is a broad consensus on the clinical practice guideline (CPG) statements used related to anticoagulation indication, patient follow-up, anticoagulation therapy complications, COVID-19 management and prevention, and cardiac interventions.

## 1. Introduction

TAF is the most common cardiac arrhythmia in adults [1,2] and can be either asymptomatic or very disabling [3]. During the last few decades, its prevalence has increased to between 2% and 4% of the adult population, which represents about 46.3-million people worldwide, and it is expected to increase by 2.3 million in the coming decades largely owing to the extended longevity of the general population. The reasons for the observed increase in AF prevalence are not completely understood but may include enhanced detection, rising incidence, improved survival in patients with cardiovascular (CV) conditions predisposing to atrial fibrillation, and greater survival following atrial fibrillation onset [4].

Atrial fibrillation (AF) is associated with an increased risk of cerebral and peripheral arterial thromboembolic incidents [5], thus representing a major public health problem with high comorbidity, an increased relapse and mortality risk, and soaring health care costs [4]. Appropriate preventive treatment for people with ischemic risks is, therefore, essential [1,6]. The simple Atrial Fibrillation Better Care (ABC) holistic pathway (“A” Anticoagulation/Avoid stroke; “B” Better symptom management; “C” Cardiovascular and Comorbidity optimization) is the framework for the general care of AF patients. It has been significantly associated with a lower risk of all causes of death, adverse cardiovascular events, hospitalizations, and lower health-related costs [7,8,9].

The prevention of stroke and systemic thromboembolism with OAC is the cornerstone for the management of atrial fibrillation. Prior to 2009, Vitamin K antagonists (VKA) such as warfarin were the drugs of choice with a high efficacy and known safety profile [10,11]. Since 2009, four DOACs, including one direct thrombin inhibitor (dabigatran etexilate) and three factor Xa inhibitors (apixaban, edoxaban, and rivaroxaban) [12], have been compared with VKA therapy for stroke prevention in non-valvular AF (NVAF) [2]. They have been shown to be at least as effective as VKA in antithrombotic prevention and treatment with a better safety profile. Indeed, they present two major advantages: (1) They are less likely to lead to hemorrhagic events, especially the most severe ones; and (2) They do not require INR monitoring as their anticoagulant effect is very stable and more independent of factors such as patients’ diets or concomitant treatment [12]. Current clinical guidelines recommend DOACs as the preferable therapy for stroke prevention in patients with NVAF [13,14,15].

Multiple risk assessment models have been developed to estimate an individual patient’s risk of stroke or systemic thromboembolism. The first model was the CHADS2 Score, which was developed in 2001 by expert consensus [2]. In 2010, Lip and colleagues published the CHA2DS2-VASc score as an update to the CHADS2 [16]. It was designed to reduce the number of patients with an intermediate risk and to better identify those who were at a low risk of thromboembolic complications [2]. A second model, called the ABC stroke risk score, has also been validated [1]. These two models have been identified as possessing the best evidence for predicting thromboembolic risk. However, in various analyses, the CHA2DS2-VASc score has been shown to have similar or modestly better predictive ability than its predecessor (the CHADS2 score) [1], which led to it being incorporated into most major guidelines as the recommended stroke risk stratification tool [2]. The decision to start a patient on an OAC should not only be based on the benefits but also on the risks (e.g., bleeding) for the individual patient. Several bleeding risk-assessment tools based on patient risk factors have been developed: HAS-BLED is a balanced tool in terms of sensitivity and specificity, whereas the European score, ABC, and mOBRI are high-sensitivity tools, and ORBIT, ATRIA, Shireman, and GARFIELD-AF are high-specificity tools [17]. The HAS-BLED tool has been validated in several clinical trials and is currently the most frequently used tool to screen for the risk of bleeding [18]. This tool incorporates the following risk factors: hypertension, abnormal renal/liver function, stroke, bleeding history or predisposition, labile international normalized ratio, elderly patients (>65 years), and drugs/alcohol concomitantly.

Prevention of the ischaemic risk with OACs is widely recommended by different global clinical practice guidelines, and recommendations have been made on the management of patients on OACs in different, complex clinical situations [1]. However, there are currently no clinical practice guidelines or recommendation documents on the optimal management of OACs in patients with NVAF that specifically address and adapt to the Central American and Caribbean context.

In order to meet this need, a consensus based on scientific evidence, and the opinion of experts, has been proposed to respond to those doubts that may arise in the management of OACs as a preventive treatment of thrombotic events related to NVAF in this geographical area.

## 2. Materials and Methods

This study is based on a review of clinical practice guidelines (CPG) and a two-round *Delphi*-type consensus survey (See Figure 1).

A scientific committee was formed, consisting of 19 cardiologists with significant experience in the management of patients with NVAF from El Salvador, Guatemala, Honduras, Panamá, Nicaragua, Costa Rica, Dominican Republic, and Spain. This committee was responsible for the decision-making, including establishing the topics to answer in the study, approving the methodology, identifying the bibliography, proposing the panelists, developing the *Delphi* statements, and analyzing the results.

A research protocol was developed that described the objectives and methodology of the project, as well as the criteria and requirements for the selection of survey respondents. The scientific committee validated the protocol and developed nine clinical questions following the PICO method (Patient, Intervention, Comparison, Outcomes) [19].

In April 2022, a review of CPGs on NVAF management was conducted, and five guidelines were selected to meet the consensus needs using the following criteria: title, national or international guideline, country, year of publication, and relevance for the project (Table 1). Twenty-eight recommendations that responded to the clinical questions established were extracted from the CPG.

The recommendations were analyzed in a meeting with the scientific committee in May 2022, applying a structured process, in order to categorize them in terms of accordance, utility, relevance, validity, and feasibility for Central America and the Caribbean, based on an ADOLOPMENT approach [26,27]. This approach combines the advantages of adoption, adaption, and de novo development of guidelines and facilitates structured interaction and deliberation with experts during meetings saving important resources [27] (See Figure 2).

During the meeting, the experts discussed the relevance to adopt, adapt, or contextualize the international recommendations, specifically in the Central America and Caribbean region. They also proposed new topics not covered in the initial draft of the project. Moreover, two extra clinical questions were also considered. To answer these new questions, a review of CPGs or consensus documents was conducted, two new CPGs were selected and analyzed, and 11 new recommendations were extracted. The scientific committee ratified the statements to be included in the *Delphi*-type questionnaire through an online survey (Table 2, Table 3, Table 4, Table 5, Table 6 and Table 7).

A panel of 30 participants was selected from a representative series of hospitals and geographic areas from Central America and the Caribbean. Panelists had to meet the following criteria: (1) Experienced in the management of patients with NVAF; (2) Leadership within the medical community; (3) Attending a reasonable number of patients with this disease. The panelists participated in the consensus process to validate the statements through a two-round *Delphi*-like methodology [28]. The statement questionnaire was designed to be completed online, with a voting system to indicate the level of agreement and fields for the panelists’ comments. It was uploaded on an online platform that also offered access to the CPGs that were used for the questionnaire development. The level of agreement was assessed on a four-point Likert scale (1: Strongly disagree; 2: Disagree; 3: Agree; 4: Strongly agree). Consensus was pre-defined as ≥80% of all respondents rating their agreement as 3 or 4, and unanimous consensus as 100% agreement (all participants voting 4). After the first round, the results and comments of the panelists were analyzed, and statements that reached consensus and those that did not require modification were not submitted to the second round.

In addition, in order to understand the need to adapt and contextualize the international recommendations and the requirements for their implementation in the Central American and Caribbean area, it was decided to ask four open questions during the second round.

## 3. Results

Related to the ADOLOPMENT-like process, from the initial 28 recommendations, 17 of them were adopted (*statement accepted*), two were contextualized (*statement’s evidence is not modified but some local information is added*), one was adapted (*statement is modified following local evidence*), and eight were rejected by the experts. The 11 recommendations formulated to answer new questions were all adopted. Finally, the statements questionnaire included 31 recommendations (Table 2, Table 3, Table 4, Table 5, Table 6 and Table 7).

For the *Delphi* survey, all 30 invited panelists participated in the two rounds. Among them, six were from Guatemala, three from El Salvador, three from Honduras, three from Nicaragua, four from Costa Rica, two from Panama, seven from Dominican Republic, and two from Puerto Rico. Twenty-seven out of the 30 were cardiologists, two were neurologists, and one was a geriatric specialist. Half of the experts reported visiting between 25 and 50 patients a month, 40% less than 25 patients a month, and only 10% reported visiting more than 50 patients a month.

In the first round, 31 recommendations were evaluated and 30 reached consensus, of which 10 were unanimously agreed upon. The only recommendation presenting discrepancy [*“For patients at “low stroke risk” (CHA2DS2-VASc score = 0 in men, or 1 in women) antithrombotic therapy should not be offered”*] was re-submitted unchanged in the second *Delphi* round as the Scientific Group did not consider it necessary to rephrase it. The results of both rounds can be seen in Table 2, Table 3, Table 4, Table 5, Table 6 and Table 7.

Regarding the additional questions, the first one referred to the considerations that, according to the panelists, should be considered for the implementation and the follow-up of the *Delphi* statements as recommendations in their country. The primary considerations were the necessity to implement medical training on AF and anticoagulation and the transmission of the latest scientific evidence to all the professionals implicated in the healthcare process. Then, panelists also considered the importance of a better access to practical information about DOACs (e.g., situations of use, early treatment data or impact on morbidity, mortality, and cost) in order to optimize therapeutic decision-making, and highlighted the importance of the economic aspects of the area. (Appendix A).

With the second question, the panelists were asked about the actions to implement in order to facilitate the use and access to DOACs in this area. More than half emphasized the need to act on the economics and prices of DOACs, arguing that their cost is one of the main barriers to their access and use. The other proposed actions were mainly related to information and training on the use of DOACs, and more particularly to the need to inform patients about the risks and benefits of DOACs throughout informative campaigns (Appendix A).

The third question was about the availability of the panelists for monitoring the implementation grade of the consensus recommendations in their country in a particular time frame (about 1 year), and whether they would be interested in collaborating. Most of the experts answered that they would be available, even if some of them pointed out the lack of time and some obligations as reasons for not being able to commit.

Finally, the experts were asked whether they thought it was necessary to implement a monographic consultation dedicated to anticoagulation for the follow-up of patients. A great majority responded positively. They were also asked about the optimal frequency of this intervention. The responses were disparate, with various ranges of frequency (from every month up to 2 years) and depending on whether or not it was the first year of treatment.

## 4. Discussion

Our study focused on defining the optimal management of OACs in patients with NVAF in Central America and the Caribbean area through the adaptation or contextualization of recommendations from international and recognized CPGs. The recommendations were validated through a two-round *Delphi*-type consensus with experienced cardiologists, neurologists, and geriatricians from the eight countries of the area [1,20,21,22,23,24,25].

The selection of an anticoagulant agent should be based on shared decision-making that considers risk factors, cost, tolerability, patient preference, potential for drug interactions, and other clinical characteristics, including time in the INR therapeutic range if the patient has been on VKA [22]. As it is shown in the results, DOACs, if they are available, are the recommended drugs for the management of patients with NVAF. The results of our study not only showed agreement in the statements related to anticoagulation indication but also in most of the statements regarding patient follow-up, anticoagulation therapy complications, COVID-19 management and prevention, and cardiac interventions.

The scientific committee included information to contextualize about cost and access to DOACs or to prothrombin complex concentrates, and it considered VKA in absence of DOACs as a valid first-line therapeutic option. The ADOLOPMENT approach, which consists of adapting and contextualizing recommendations, may have had a positive effect among the panelists on achieving consensus in some of the statements. This approach is considered a rigorous, valid, and reproducible alternative methodology for obtaining clinical recommendations at a local level in a lesser time and using fewer resources. Indeed, the use of guidelines developed in other settings may be inappropriate because of different contextual factors such as acceptability or feasibility. There are some limitations to their adaptation, as recommendations are modified to reflect these contextual factors. However, it has been hypothesized that adapting the guidelines to the local setting is expected to improve their uptake and implementation [26,27,29].

Only one of the 31 recommendations did not reach consensus and showed discrepancies between panelists. The discrepant recommendation was related to the management of patients with low stroke risk (CHA2DS2-VASC = 0 in men or 1 in women) and the need of antithrombotic therapy. In the first *Delphi* round, only 60% of the experts agreed on not offering antithrombotic therapy in this group of patients. Although the percentage of agreement increased to 70% in the second round, it did not meet the consensus level (80%), and the statement remained discrepant.

The scientific committee considered that atrial fibrillation, by itself, is not a reason to initiate anticoagulation therapy, and it must be considered that in low-risk stroke patients anticoagulation risks (mainly bleeding) outweigh the benefits. However, some of the panelists considered that the indication of anticoagulant therapy must be broader. Discrepancy could be explained because there is less agreement on whether to recommend anticoagulation or antiagregation in low-risk patients among reference guideline recommendations. For men with a CHA2DS2-VASC score of 0 and 1 in women, ACCP and ESC guidelines [14,15] recommend omitting antithrombotic therapy, but AHA/ACC/HRS [13] makes a weaker recommendation, stating that it is reasonable to omit anticoagulant therapy. Among panelists’ comments, it stands out that some of them consider patients with atrial fibrillation candidates to anticoagulant therapy if there is no absolute contradiction, regardless of the CHA2DS2-VASC score. Other experts noted that this decision should be individualized, arguing that the CHA2DS2-VASC score does not include pro-coagulant factors such as AF burden or left atrial volume index.

Results of the open-ended questions formulated to panelists are aligned with the process followed by the scientific committee to adapt and contextualize recommendations. In regard to the implementation of recommendations, panelists have pointed out the need to dispose of more practical information about DOAC, to develop local medical training projects, and to implement initiatives for lowering the DOAC’s costs and improving their local access. Experts consider that elevated costs are one of the main limitations to promote the use and access to DOACs in the Central America and Caribbean area and actions must be considered in order to reduce them. VKAs have a lower cost but enable poor anticoagulation control, which has a strong impact on health loss and on increased health system expenses [30]. Reference-pricing policies and the use of generic drugs could lead to decreases in drug prices and to increases in utilization of targeted medications, while also reducing payer and patient expenditures [31,32]. Generic drugs are considered to have the same qualitative and quantitative composition in active substance and the same pharmaceutical form, and whose bioequivalence with the reference drug has been demonstrated by appropriate bioavailability studies [33].

This consensus document can serve as an adapted local guide for the management of OACs in patients with NVAF in the Central American and Caribbean area. In the near future, it will be necessary to readapt it due to the use artificial intelligence (AI) and machine learning in cardiology [34]. AI applications have shown effectiveness in managing AF, aiding in risk assessment beyond CHA2DS2-VASc and HAS-BLED, diagnosis, treatment selection, and remote monitoring. Despite these challenges, such as the need for extensive, high-quality data and ethical considerations, for example, preparing for the AI era is essential for physicians to enhance patient outcomes in chronic diseases.

The design of this study has some limitations inherent to the chosen methodology. Although it was conducted using a robust, well-known, rigorous methodology based on the *Delphi* technique, it only provides us with qualitative information on the degree of agreement among the panelists based on the available evidence, as well as their clinical practice and experience.

## 5. Conclusions

The present *Delphi*-like study assessed expert opinions in a wide variety of contextualized recommendations for the optimal management of DOACs in patients with NVAF. There is a broad consensus on the CPG statements used related to anticoagulation indication, patient follow-up, anticoagulation therapy complications, COVID-19 management and prevention, and cardiac interventions. Considering this, this consensus document can serve as an adapted local guide for the management of these patients.

The results manifest that DOACS are recommended over VKAs. It is important to individualize treatment according to patient’s thrombotic and bleeding risk and to select the best therapeutic strategy conditioned by the level of access to medicines and the clinical context of the patient. As cost and access are important limitation factors, efforts must be made to allow for a better access to DOACs as first line-treatment options in patients with NVAF from Central America and the Caribbean.

## Figures and Tables

**Figure 1 jcm-13-00314-f001:**
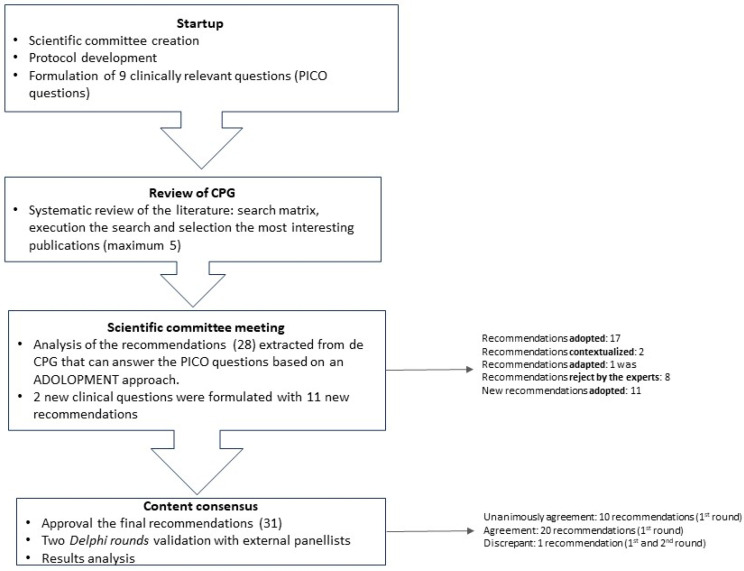
Methodology flowchart.

**Figure 2 jcm-13-00314-f002:**
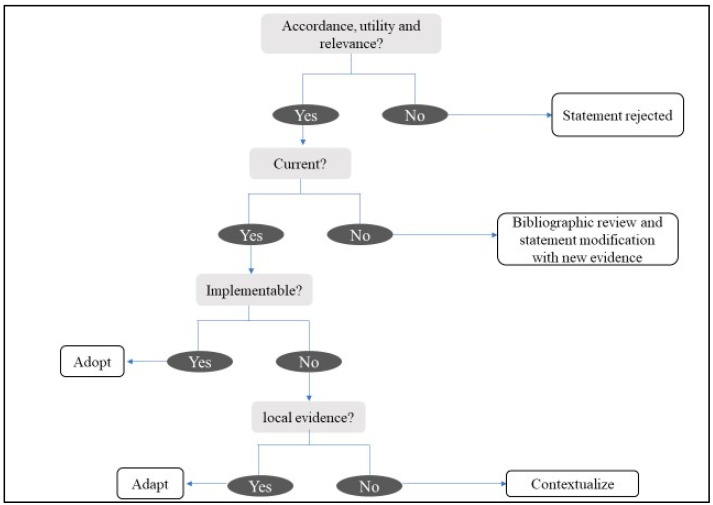
Analysis and evaluation of recommendations.

**Table 1 jcm-13-00314-t001:** CPG selection.

Title [*Year*]	Journal	Region/Country	Year
*First CPG selection*			
“ESC Guidelines for the diagnosis and management of atrial fibrillation developed in collaboration with the European Association for Cardio-Thoracic Surgery (EACTS). 2021” [1]	Eur Heart J.	Europe	2021
“Joint European consensus document on the management of antithrombotic therapy in atrial fibrillation patients presenting with acute coronary syndrome and/or undergoing percutaneous cardiovascular interventions. 2019” [20]	ESC Scientific Document Group, Europace	Europe	2019
“ESC guidance for the diagnosis and management of cardiovascular disease during the COVID-19 pandemic. 2022” [21]	Eur Heart J.	Europe	2022
“Update to the 2016 ACC/AHA Clinical Performance and Quality Measures for Adults With Atrial Fibrillation or Atrial Flutter. 2021” [22]	J Am Coll Cardiol.	America	2021
“ACC Expert Consensus Decision Pathway on Management of Bleeding in Patients on Oral Anticoagulants. 2020” [23]	J Am Coll Cardiol.	America	2020
*Second CPG selection*			
“Perioperative and Periprocedural Management of Antithrombotic Therapy: Consensus Document of SEC, SEDAR, SEACV, SECTCV, AEC, SECPRE, SEPD, SEGO, SEHH, SETH, SEMERGEN, SEMFYC, SEMG, SEMICYUC, SEMI, SEMES, SEPAR, SENEC, SEO, SEPA, SERVEI, SECOT, and AEU. 2018” [24]	Rev Esp Cardiol.	Spain	2018
“2017 ACC Expert Consensus Decision Pathway for Periprocedural Management of Anticoagulation in Patients With Nonvalvular Atrial Fibrillation. 2017” [25]	J Am Coll Cardiol.	America	2017

**Table 2 jcm-13-00314-t002:** First-section consensus statements and level of agreement achieved during the two Delphi rounds.

Section 1—Oral Anticoagulation (OAC) Indication				
Statement		Level of Agreement 1st Round	Level of Agreement 2nd Round	Final Result
ADOLOPMENT-like Process				
**PICO 1: Which profile of patient with NVAF have indications of OAC (VKA or DOAC) for the prevention of stroke?**				
1.1. “OAC is recommended for stroke prevention in AF patients with CHA2DS2-VASc score ≥ two in men or ≥ three in women” [1].	Adopted	93.33%	-	Consensus
1.2. “OAC should be considered for stroke prevention in AF patients with a CHA2DS2-VASc score of one in men or two in women. Treatment should be individualized based on net clinical benefit and consideration of patient values and preferences” [1].	Adopted	90.00%	-	Consensus
1.3. “For patients at “low stroke risk” (CHA2DS2-VASc score = 0 in men, or 1 in women) antithrombotic therapy should not be offered” [1].	Adopted	60.00%	70.00%	Discrepancy
**PICO 2: Is there any difference in the indication of OAC in patients with NVAF, according to whether it is paroxysmal or persistent?**				
2.1. “Selection of anticoagulant therapy should be based on the risk of thromboembolism, irrespective of whether the AF pattern is paroxysmal, persistent, or permanent” [22].	Adopted	86.67%	-	Consensus
**PICO 3: Is there any difference in the indication of OAC in patients with NVAF and in patients with atrial flutter?**				
3.1. “For patients with atrial flutter, anticoagulant therapy is recommended according to the same risk profile used for AF” [22].	Adopted	93.33%	-	Consensus
**PICO 4: What kind of OAC (VKA or DOAC) is best for preventing stroke in patients with NVAF?**				
4.1 “DOACs are recommended in preference to VKAs (excluding patients with mechanical heart valves or moderate-to-severe mitral stenosis) for stroke prevention in AF patients who are eligible for OAC. In the absence of DOACs, warfarin may be a valid option for stroke prevention in AF patients, only when there is a correct control of INR (control frequency and therapeutic range)” [22].	Adapted	93.33%	-	Consensus

DOAC: Direct oral anticoagulants; NVAF: Non-valvular atrial fibrillation; OAC: Oral anticoagulant; VKA: Vitamin K antagonist.

**Table 3 jcm-13-00314-t003:** Second-section consensus statements and level of agreement achieved during the two Delphi rounds.

Section 2—Anticoagulated-Patient Monitoring				
Statement		Level of Agreement 1st Round	Level of Agreement 2nd Round	Final Result
ADOLOPMENT-like Process				
**PICO 5: What is the treatment follow-up for patients with NVAF treated with OACs?**				
5.1. “If a VKA is used, a target INR of 2.0–3.0 is recommended, with individual TTR ≥ 70% [1]. Among patients treated with VKA, the INR should be determined at least weekly during initiation of anticoagulant therapy and at least monthly when anticoagulation (INR in range) is stable [22]. In patients on VKA with low time in INR therapeutic range, recommended options are: - Switching to a DOAC but ensuring good adherence and persistence with therapy; [1] or - Efforts to improve TTR (e.g., education/counselling and more frequent INR checks)” [1].	Adopted	100.00%	-	Unanimity
5.2. “In AF patients, stroke risk must be assessed using the CHA2DS2-VASc score, and bleeding risk should be assessed using the HAS-BLED score [20]. - Stroke- and bleeding-risk stratification are dynamic processes and must be performed at regular intervals. - Every effort should be made to address modifiable bleeding risk factors at every patient contact. - Established bleeding scores, e.g., HAS-BLED, should be used to draw attention to modifiable bleeding risk factors and to identify the patients for earlier review and follow-up.”	Adopted	100.00%	-	Unanimity
5.3. “For bleeding risk assessment, a formal structured risk-score-based bleeding risk assessment is recommended to help identify non-modifiable and address modifiable bleeding risk factors in all AF patients, and to identify patients potentially at high risk of bleeding who should be scheduled for early and more frequent clinical review and follow-up” [1].	Adopted	96.67%	-	Consensus
5.4. “For a formal risk-score-based assessment of bleeding risk, the HAS-BLED score should be considered to help address modifiable bleeding risk factors and to identify patients at high risk of bleeding (HAS-BLED score ≥ 3) for early and more frequent clinical review and follow-up”.	Adopted	96.67%	-	Consensus

DOAC: Direct oral anticoagulants; INR: International normalized ratio; NVAF: Non-valvular atrial fibrillation; OAC: Oral anticoagulant; TTR: Time in therapeutic range; VKA: Vitamin K antagonist.

**Table 4 jcm-13-00314-t004:** Third-section consensus statements and level of agreement achieved during the two *Delphi* rounds.

Section 3—Anticoagulation Complications.				
Statement		Level of Agreement 1st Round	Level of Agreement 2nd Round	Final Result
ADOLOPMENT-like Process				
**PICO 6: What is the therapeutic approach of patients with NVAF treated with OAC presenting bleeding complications?**				
6.1. “In an AF patient with severe active bleeding, it is recommended to [1]: - Interrupt OAC until the cause of bleeding is identified and active bleeding is resolved; and - Promptly perform specific diagnostic and treatment interventions to identify and manage the cause(s) and source(s) of bleeding”.	Adopted	100%	-	Unanimity
6.2. As long as it is available, four-factor prothrombin complex concentrates should be considered in AF patients on VKA who develop a severe bleeding complication.	Contextualized	96.67%	-	Consensus
6.3. “For patients with a non-major bleed, the writing committee does not support routine reversal of the OAC, although it is often advisable to temporarily discontinue OAC therapy until the patient is clinically stable and hemostasis has been achieved. - If it is determined that the patient does not require hospitalization, a procedure, or a transfusion, and hemostasis has been achieved, the writing committee supports continuing the OAC” [23].	Adopted	100.00%	-	Unanimity

AF: Atrial fibrillation; NVAF: Non-valvular atrial fibrillation; OAC: Oral anticoagulant; VKA: Vitamin K antagonist.

**Table 5 jcm-13-00314-t005:** Fourth-section consensus statements and level of agreement achieved during the two *Delphi* rounds.

Section 4—COVID-19.				
Statement		Level of Agreement 1st Round	Level of Agreement 2nd Round	Final Result
ADOLOPMENT-like Process				
**PICO 7: In a patient with NVAF and SARS** **-** **CoV** **-** **2 infection, what is the recommended approach related to OAC?**				
7.1. “Management of cardiac arrhythmias in patients with COVID-19: In general, the acute treatment of arrhythmias should not be significantly different from their management in non-COVID-19 patients” [21].	Adopted	96.67%	-	Consensus
7.2. “In COVID-19 patients with an indication for oral anticoagulant therapy, renal, and liver function, drug-drug interactions between oral anticoagulant and COVID-19 therapies should be considered to minimize the risk of bleeding or thromboembolic complications” [21].	Adopted	96.67%	-	Consensus
7.3. In DOAC-eligible patients (e.g., those without mechanical prosthetic heart valves, moderate-to-severe mitral stenosis, or antiphospholipid syndrome), DOACs are preferred over Vitamin K antagonists (VKAs) as long as local restrictions and resources allow it, owing to their better safety and fixed dosing without the need for laboratory monitoring of anticoagulant effect, notwithstanding the importance of proper DOAC dosing and adherence to treatment [21].	Contextualized	96.67%	-	Consensus
7.4. “Whereas apixaban, rivaroxaban, or edoxaban can be provided as oral solutions or crushed tablets (via enteral tubes), severely ill COVID-19 patients may be switched to parenteral anticoagulation, which has no clinically relevant drug-drug interactions with COVID-19 therapies (with the exception of azithromycin, which should not be coadministered with UFH)” [21].	Adopted	93.33%	-	Consensus
**PICO 8: Is there any contradiction to COVID** **-** **19 vaccination in patients with NVAF treated with OACs?**			**-**	
8.1. “There are very few contraindications to vaccines, and AF is not a contraindication per se” [1].	Adopted	96.67%	-	Consensus
8.2. “Intramuscular injections are considered a low bleeding-risk procedure, with little evidence contraindicating or describing complications in anticoagulated patients. Despite the fact that multiple guidelines and leaflets suggest avoiding intramuscular injections in these patients, in the face of a major pandemic such as COVID and the absence of subcutaneous vaccines, the risk of bleeding is very low, manageable, and is not considered a contraindication for its application” [24].	Adopted	100.00%	-	Unanimity

DOAC: Direct oral anticoagulants; NVAF: Non-valvular atrial fibrillation; OAC: Oral anticoagulant; UFH: Unfractionated heparin; VKA: Vitamin K antagonist.

**Table 6 jcm-13-00314-t006:** Fifth-section consensus statements and level of agreement achieved during the two *Delphi* rounds (part 1).

Section 5—Surgical Interventions.				
Statement		Level of Agreement 1st Round	Level of Agreement 2nd Round	Final Result
ADOLOPMENT-like Process				
**PICO 9: In patients with NVAF treated with OAC undergoing cardiac interventions (angioplatia, ablation), w** **hat is the therapeutic approach?**				
9.1. “For patients undergoing AF catheter ablation who have been therapeutically anticoagulated with warfarin, dabigatran, rivaroxaban, apixaban, or edoxaban, performance of the ablation procedure without OAC interruption is recommended” [1].	Adopted	93.33%		Consensus
9.2. “After AF catheter ablation, it is recommended that [1]: - Systemic anticoagulation with warfarin or a DOAC is continued for at least 2 months post ablation, and - Long-term continuation of systemic anticoagulation beyond 2 months post ablation is based on the patient’s stroke risk profile (based on CHA2Ds2-VASc score) and not on the apparent success or failure of the ablation procedure.”	Adopted	100.00%		Unanimity
9.3. “In AF patients with acute coronary syndrome undergoing an uncomplicated PCI, early cessation (≤1 week) of aspirin and continuation of dual therapy with an OAC and a P2Y12 inhibitor (preferably clopidogrel) for up to 12 months are recommended if the risk of stent thrombosis is low or if concerns about bleeding risk prevail over concerns about risk of stent thrombosis, irrespective of the type of stent used” [1].	Adopted	90.00%		Consensus
9.4. “After uncomplicated PCI, early cessation (≤1 week) of aspirin and continuation of dual therapy with OAC for up to 12 months and clopidogrel is recommended if the risk of stent thrombosis is low or if concerns about bleeding risk prevail over concerns about risk of stent thrombosis, irrespective of the type of stent used” [1].	Adopted	80.00%		Consensus
9.5. “Periprocedural management in patients undergoing Percutaneous coronary interventions (PCI) [20]: Elective PCI: - Because of the reduced risk of bleeding, VKA should not be interrupted (or bridged with heparin). - DOACS: No bridging is recommended. Emergency PCI: DOACs need not to be interrupted”.	Adopted	90.00%		Consensus
9.6. “Post-procedural management in patients with VKA undergoing elective Percutaneous coronary interventions (PCI) [20]: Antithrombotic regimen/Intensity of oral anticoagulant - In patients on triple antithrombotic therapy (TAT), INR at the lower end of therapeutic range (2.0–2.5) should be targeted, with high time in therapeutic range (TTR) (>65–70%). - With dual antithrombotic therapy (DAT), conventional therapeutic range (2.0–3.0) may be targeted with high TTR (>65–70%). Intensity of OAC during subsequent antithrombotic regime after 12 months: - Target INR should be 2.0–2.5 after withdrawal of one antiplatelet agent, with high TTR (>65–70%). Duration of TAT - Based on the risk of stent thrombosis/recurrent cardiac events and bleeding, 1-to-3–6 months should be selected. Especially with acute coronary syndrome patients, ideally try to keep TAT to 6 months, but shorten to 3 months if high bleeding risk (e.g., HAS-BLED ≥ 3).”	Adopted	100.00%		Unanimity
9.7. “Post-procedural management in patients with DOAC undergoing elective PCI [20]: Antithrombotic regimen/Intensity of oral anticoagulant Low-dose dabigatran of 110 mg bid, full-dose apixaban of 5 mg bid, and edoxaban 60 mg of od should be selected to optimize risk-benefit ratio if part of a TAT regime. - With DAT, dabigatran of 150 mg, plus P2Y12, is preferred unless dose-reduction criteria for dabigatran are present in accordance with its label. - Reduced low-dose rivaroxaban 15 mg od rather than full dose 20 mg od may be considered to reduce the risk of bleeding. - Pending further data in the PCI setting, reduced dose of apixaban and edoxaban are only used in accordance with their respective approved labels. Intensity of OAC during subsequent antithrombotic regime after 12 months: - After withdrawal of one antiplatelet agent, full-dose apixaban of 5 mg bid and edoxaban of 60 mg od should be used, whereas reduced dose of rivaroxaban of 15 mg od should be replaced by full-dose of 20 mg od if creatinine clearance is ≥50 mL/min. - Decision on whether or not to increase ongoing dabigatran of 110 mg bid-to-150 mg bid should be left at the discretion of the attending physician based on the individual risk of stroke and bleeding and the goal of antithrombotic therapy.”	Adopted	96.67%		Consensus

AF: Atrial fibrillation; DOAC: Direct-oral anticoagulants; INR: International normalized ratio; NVAF: Non-valvular atrial fibrillation; OAC: Oral anticoagulant; PCI: Percutaneous coronary intervention; TAT: Triple antithrombotic therapy; TTR: Time in therapeutic range; UFH: Unfractionated heparin; VKA: Vitamin K antagonist, DAT: Dual antithrombotic therapy.

**Table 7 jcm-13-00314-t007:** Fifth-section consensus statements and level of agreement achieved during the two *Delphi* rounds (Part 2).

Section 5—Surgical Interventions.				
Statement		Level of Agreement 1st Round	Level of Agreement 2nd Round	Final Result
ADOLOPMENT-like Process				
**PICO 10: In patients with NVAF treated with OAC undergoing elective surgery, what is the therapeutic approach considering bleeding risk?**				
10.1. “For patients using VKA [23] Do not interrupt therapy with a VKA in: - Patients undergoing procedures with (1) Not clinically important or low bleed risk; and (2) Absence of patient-related factor(s) that increase the risk of bleeding. Interrupt therapy with a VKA in: - Patients undergoing procedures with intermediate or high bleed risk, or - Patients undergoing procedures with uncertain bleed risk and the presence of patient-related factor(s) that increase the risk of bleeding. Consider interrupting a VKA based on both clinical judgment and consultation with the proceduralist in: - Patients undergoing procedures with (1) Not clinically important or low bleed risk; and (2) The presence of patient-related factor(s) that increase the risk of bleeding, or - Patients undergoing procedures with (1) Uncertain bleed risk; and (2) The absence of patient-related factor(s) that increase the risk of bleeding.”	Adopted	90.00%		Consensus
10.2. “For patients using direct oral anticoagulants (DOAC) [23] Do not interrupt therapy with a DOAC in: - Patients undergoing procedures with (1) No clinically important or low bleed risk; and (2) Absence of patient-related factor(s) that increase the risk of bleeding. Interrupt therapy for intermediate, high, or uncertain bleed-risk procedures in: - Patients treated with any of the approved DOACs for a duration based on the estimated CrCl”	Adopted	90.00%		Consensus
10.3. “Bridging therapy in patients on a VKA and risk for thromboembolism [23]: For patients with very low risk for thromboembolism, there is no need of bridging therapy. For patients who are at low risk for thromboembolism (<5%/year), with a CHA2DS2-VASc score ≤ 4 and no prior history of ischemic stroke, TIA, or SE, discontinue the VKA prior to the procedure and resume as discussed in the following text, without bridging. For determining appropriateness for bridging in those on a VKA at moderate risk for thromboembolism (5% to 10%/year) with a CHA2DS2-VASc score of 5 to 6, or history of prior ischemic stroke, TIA, or peripheral arterial embolism (3 months previously). Determine the patient’s bleed risk to determine the appropriateness of bridging therapy. - If increased risk of bleeding, interruption of the VKA without bridging is recommended. - If no significant bleed risk: ▪ In patients with prior stroke, TIA, or SE, consider use of a parenteral anticoagulant for periprocedural bridging (use clinical judgment, likely bridge); ▪ In patients with no prior stroke, TIA, or SE, the use of a parenteral anticoagulant for periprocedural bridging is not advised (use clinical judgment, likely do not bridge.) For patients who are at high risk of stroke or systemic embolism (>10% per year) with a CHA2DS2-VASc score of 7-to-9 or recent (within 3 months) ischemic stroke, TIA, or SE, parenteral bridging anticoagulation should be considered.”	Adopted	96.67%		Consensus
10.4. “For restarting anticoagulation post-procedure [23]: Ensure procedural site hemostasis. Consider bleeding consequences, especially with high bleed-risk procedures such as open cardiac surgical, intracranial, or spinal procedures. Consider patient-specific factors that may predispose the patient to bleeding complications (e.g., bleeding diathesis, platelet dysfunction, antiplatelet medications).”	Adopted	100%	-	Unanimity

CrCl: Creatinine clearance; DOAC: Direct-oral anticoagulants; NVAF: Non-valvular atrial fibrillation; OAC: Oral anticoagulant; SE: Systemic embolism; TIA: Transient ischaemic attack; VKA: Vitamin K antagonist.

## Data Availability

No new data were created or analyzed in this study. Data sharing is not applicable to this article.

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
