# Peer review of "Central American and Caribbean Consensus Document for the Optimal Management of Oral Anticoagulation in Patients with Non-Valvular Atrial Fibrillation Endorsed by the Central American and Caribbean Society of Arterial Hypertension and Cardiovascular Prevention"

_jcm, 2024, doi:10.3390/jcm13020314_

Round 1
Reviewer 1 Report
Comments and Suggestions for Authors
The study assessed expert opinion in a wide variety of contextualized recommendations for the optimal management of OACs in patients with NVAF in Central American and the Caribbean area. A few minor revisions should be considered.
1. In Table1, the content of "1.4" was the same with that of "1.1".
2. In Table 4 "8.1", the "FA" was a slip of the pen?
3. In Table 5 "9.7", the setence of "Antithrombotic regimen/Intensity of oral anticoagulant" was repeated twice.
Author Response
1. Summary |
|
|
Thank you very much for taking the time to review this manuscript. Please find the detailed responses below and the corresponding revisions/corrections highlighted in the resubmitted files.
|
||
2. Point-by-point response to Comments and Suggestions for Authors |
||
Comments 1: In Table 1, the content of "1.4" was the same with that of "1.1". |
||
Response 1: Thank you for pointing this out. We have reviewed the results and notified that it was a mistake. We have delated the sentence 1.4. |
||
Comments 2: 8.1", the "FA" was a slip of the pen? Response 2: Sure, it was typo mistake. We have corrected it using the correct acronym.
Comments 3: In Table 5 "9.7", the sentence of "Antithrombotic regimen/Intensity of oral anticoagulant" was repeated twice. Response 3: It was another typo mistake. We have delated the repeated sentence.
|
Thank you for considering the changes and being able to proceed with the publication process.
Kind regards,
The authors
Reviewer 2 Report
Comments and Suggestions for Authors
1.The authors state that a “consensus project was performed based on a systematic review of the literature”. However, it is not clear which part of the manuscript is a systematic review. If they conducted a systematic review include the search strategy and platforms used at least in the supplementary file.
2. Table 1 should include more details (year, country etc) of the CPG selection and can be simplified by using different columns for different details.
3. Figures 1 and 2 is not legible.
4. There are many typo errors (eg Line 5, 85 etc)
5. Authors should try to give a more scientific explanation for the statement with a lower level of agreement.
6. Is there any regional-wise difference in the management of OAC in NVAF patients?
7. Conclusion of the study should state what is the overall finding of the current study and how the study will be useful in the management of OAC in NVAF patients.
Author Response
1. Summary |
|
|
||||||||||||||||
Thank you very much for taking the time to review this manuscript. Please find the detailed responses below and the corresponding revisions/corrections highlighted in the re-submitted files.
|
||||||||||||||||||
2. Questions for General Evaluation |
Reviewer’s Evaluation |
Response and Revisions |
||||||||||||||||
Does the introduction provide sufficient background and include all relevant references? |
Yes |
|
||||||||||||||||
Are all the cited references relevant to the research? |
Can be improved |
We consider that all the references are relevant. |
||||||||||||||||
Is the research design appropriate? |
Yes |
|
||||||||||||||||
Are the methods adequately described? |
Can be improved |
In order to improve clarity of the methods we included a figure to explain the process.
|
||||||||||||||||
Are the results clearly presented? |
Can be improved |
Results, as indicated in the text, are presented in the tables Tables 2, 3, 4, 5, 6 and 7. In order to not repeat statements we decided to include %of agreement (results) in the same table where Delphi-type questionnaire is presented (methodology section).
|
||||||||||||||||
Are the conclusions supported by the results? |
Must be improved/Not applicable
|
It will be responded in the point-by-point response.
|
||||||||||||||||
3. Point-by-point response to Comments and Suggestions for Authors |
||||||||||||||||||
Comments 1: The authors state that a “consensus project was performed based on a systematic review of the literature”. However, it is not clear which part of the manuscript is a systematic review. If they conducted a systematic review include the search strategy and platforms used at least in the supplementary file. |
||||||||||||||||||
Response 1: Thank you for pointing this out. Therefore, we have included in supplementary file the search strategy and platforms used as well as other relevant filters:
Sources of information
Search strategy, specific filters, key keywords
Comments 2: The table 1 should include more details (year, country etc) of the CPG selection and can be simplified by using different columns for different details. |
||||||||||||||||||
|
||||||||||||||||||
Response 2: We have, accordingly, included in table 1 the year of publication and the region/country from the CPG.
Comment 3: Figures 1 and 2 is not legible. Response 3: We have provided a new version from figures 1 and 2 with more quality and therefore more legible.
Comment 4: There are many typo errors (eg Line 5, 85 etc) Response 4: These errors have been corrected and also a review of the whole text have been don.
Comment 5: Authors should try to give a more scientific explanation for the statement with a lower level of agreement.
Response 5: This information is provided in the discussion section and the text is: Only one of the 31 recommendations did not reach consensus and showed discrepancies between panelists. The discrepant recommendation was related to the management of patients with low stroke risk (CHA2DS2-VASC = 0 in men, or 1 in women) and the need of antithrombotic therapy. In the first Delphi round, only 60% of the experts agreed on not offering antithrombotic therapy in this group of patients. Although the percentage of agreement has increased to 70% in the second round, it did not meet the consensus level (80%), and the statement remained discrepant. The scientific committee considered that atrial fibrillation, by itself, is not a reason to initiate an anticoagulation therapy and it must be considered that in low-risk stroke patients anticoagulation risks (mainly bleeding) outweigh the benefits. However, some of the panelists considered that the indication of anticoagulant therapy must be broader. Discrepancy could be explained because there is less agreement on whether to recommend anticoagulation or antiagregation in low-risk patients among reference guideline recommendations. For men with a CHA2DS2-VASC score of 0 and 1 in women, ACCP and ESC guidelines [14, 15] recommend omitting antithrombotic ther-apy, but AHA/ACC/HRS [13] makes a weaker recommendation, stating that it is rea-sonable to omit anticoagulant therapy. Among panelists´ comments, it stands out that some of them consider patients with atrial fibrillation candidates to anticoagulant therapy, if there is no absolute contradiction, regardless of the CHA2DS2-VASC score. Other experts noted that this decision should be individualized, arguing that the CHA2DS2-VASC score does not include pro-coagulant factors, like AF burden or left atrial volume index.
Comment 6: Is there any regional-wise difference in the management of OAC in NVAF patients? Response 6: Knowing the regional-wise difference in the management of OAC in NVAF patients was not the main objective of the study. Currently, there are no clinical practice guidelines or recommendations documents on the optimal management of OACs in patients with NVAF that specifically address and adapt to the Central American and Caribbean context. In the second Delphi round, to understand the need to adapt and contextualize the international recommendations and the requirements for their implementation in the Central American and Caribbean area, it was decided to ask some open questions during the second round. Taken into account of the results of these questions, some of them summarized in supplementary field (Figure S1. Considerations that, according to the panelists, should be taken into account for the implementation and the follow-up of the Delphi statements as recommendations in their country), there are differences in cost and availability of OAC among countries.
Comment 7: Conclusion of the study should state what is the overall finding of the current study and how the study will be useful in the management of OAC in NVAF patients.
Response 7: Overall finding of the study is there is a broad consensus among experts from different countries from Central America and Caribbean area on the CPG statements used related to anticoagulation indication, patient follow-up, anticoagulation therapy complications, COVID-19 management and prevention, and cardiac interventions. In order to improve conclusions, we have added this sentence: “, this consensus document can serve as an adapted local guide for the management of patients with NVAF”.
|
Thank you for considering the changes and being able to proceed with the publication process.
Kind regards,
The authors
Reviewer 3 Report
Comments and Suggestions for Authors
I have reviewed the manuscript entitled 'Central American and Caribbean Consensus Document for the optimal management of oral anticoagulation in patients with non-valvular atrial fibrillation endorsed by the Central American and Caribbean Society of Arterial Hypertension and Cardiovascular Prevention'.
The manuscript is well-written and described its hypothesis very well, however several issues should be checked before further evaluation. First, the continuance of anticoagulants despite successful ablation procedure should be mentioned in order to mention the importance of anticoagulation in every long-persistent AF patients. Please consider citing 'Comparison of catheter ablation and medical therapy for atrial fibrillation in heart failure patients: A meta-analysis of randomized controlled trials'.
Chadsvasc and Hasbled scores are very important according toı current evidence. However we should have more precise scores for embolism and bleeding. Please add a short section to the discussion citing 'The Role of Artificial Intelligence in Coronary Artery Disease and Atrial Fibrillation' . AI systems will be an important option to start anticoagulation in a very near future. This should be mention in this important paper.
Comments on the Quality of English LanguageI have reviewed the manuscript entitled 'Central American and Caribbean Consensus Document for the optimal management of oral anticoagulation in patients with non-valvular atrial fibrillation endorsed by the Central American and Caribbean Society of Arterial Hypertension and Cardiovascular Prevention'.
The manuscript is well-written and described its hypothesis very well, however several issues should be checked before further evaluation. First, the continuance of anticoagulants despite successful ablation procedure should be mentioned in order to mention the importance of anticoagulation in every long-persistent AF patients. Please consider citing 'Comparison of catheter ablation and medical therapy for atrial fibrillation in heart failure patients: A meta-analysis of randomized controlled trials'.
Chadsvasc and Hasbled scores are very important according toı current evidence. However we should have more precise scores for embolism and bleeding. Please add a short section to the discussion citing 'The Role of Artificial Intelligence in Coronary Artery Disease and Atrial Fibrillation' . AI systems will be an important option to start anticoagulation in a very near future. This should be mention in this important paper.
Author Response
1. Summary |
|
|
Thank you very much for taking the time to review this manuscript. Please find the detailed responses below and the corresponding revisions/corrections highlighted in the re-submitted files.
|
||
2. Questions for General Evaluation |
Reviewer’s Evaluation |
Response and Revisions |
Does the introduction provide sufficient background and include all relevant references? |
Yes |
|
Are all the cited references relevant to the research? |
Can be improved |
We consider that all the references are relevant. |
Is the research design appropriate? |
Yes |
|
Are the methods adequately described? |
Yes |
|
Are the results clearly presented? |
Yes |
|
Are the conclusions supported by the results?
|
Can be improved |
|
3. Point-by-point response to Comments and Suggestions for Authors |
||
Comments 1: The manuscript is well-written and described its hypothesis very well, however several issues should be checked before further evaluation. First, the continuance of anticoagulants despite successful ablation procedure should be mentioned in order to mention the importance of anticoagulation in every long-persistent AF patient. Please consider citing 'Comparison of catheter ablation and medical therapy for atrial fibrillation in heart failure patients: A meta-analysis of randomized controlled trials'.
|
||
Response 1: Thank you for these accurate comments for the management of patients after an ablation procedure but we consider that this point is not necessary to be included specifically in the text because it falls outside the scope of the study (The objective of the publication is to provide a consensus document on the optimal management of OACs in patients with NVAF that specifically address and adapt to the Central American and Caribbean context). On the other hand, we consider that if we break down to this level, all the recommendations, the content would become too long. Comment 2: Chadsvasc and Hasbled scores are very important according toı current evidence. However, we should have more precise scores for embolism and bleeding. Please add a short section to the discussion citing 'The Role of Artificial Intelligence in Coronary Artery Disease and Atrial Fibrillation' . AI systems will be an important option to start anticoagulation in a very near future. This should be mentioned in this important paper.
|
||
Comments 2: Fully agree with this comment. We have added a short text and the corresponding reference in the discussion: This consensus document can serve as an adapted local guide for the management of OACs in patients with NVAF in Central American and Caribbean area and in the near future it will be necessary to readapt it due to the use artificial intelligence (AI) and machine learning in cardiology [34]. AI applications have shown effectiveness in managing AF, aiding in risk assessment beyond CHA2DS2-VASc and HAS-BLED, diagnosis, treatment selection, and remote monitoring. Despite these challenges, as for example the need for extensive, high-quality data and ethical considerations; preparing for the AI era is essential for physicians to enhance patient outcomes in chronic diseases. |
||
|
||
Thank you for considering the changes and being able to proceed with the publication process. If there is any other doubt, please don’t hesitate to contact us again. Kind regards,
The authors
|